# The Effectiveness of Protection and Surveillance Zones in Detecting Further African Swine Fever Outbreaks in Domestic Pigs—Experience of the Baltic States

**DOI:** 10.3390/v16030334

**Published:** 2024-02-22

**Authors:** Kristīne Lamberga, Arvo Viltrop, Imbi Nurmoja, Marius Masiulis, Paulius Bušauskas, Edvīns Oļševskis, Mārtiņš Seržants, Alberto Laddomada, Felix Ardelean, Klaus Depner

**Affiliations:** 1Food and Veterinary Service, LV 1050 Riga, Latvia; kristine.lamberga@pvd.gov.lv (K.L.); edvins.olsevskis@pvd.gov.lv (E.O.); martins.serzants@pvd.gov.lv (M.S.); 2Institute of Food Safety, Animal Health and Environment “BIOR”, LV 1076 Riga, Latvia; 3Faculty of Veterinary Medicine, Latvia University of Life Sciences and Technologies, LV 3001 Jelgava, Latvia; 4Institute of Veterinary Medicine and Animal Sciences, Estonian University of Life Sciences, 51006 Tartu, Estonia; arvo.viltrop@emu.ee; 5National Centre for Laboratory Research and Risk Assessment, 51006 Tartu, Estonia; imbi.nurmoja@labris.agri.ee; 6Veterinary Academy, Lithuanian University of Health Sciences, LT 47181 Kaunas, Lithuania; 7State Food and Veterinary Service, LT 07170 Vilnius, Lithuania; paulius.busauskas@vmvt.lt; 8Coordinator of the Better Training for Safer Food Courses on the EU Animal Health Law, 07021 Sardinia, Italy; albertolad@live.com; 9County Sanitary Veterinary Health and Food Safety Directorate, 4400067 Satu Mare, Romania; ardelean.felix-sm@ansvsa.ro; 10Friedrich-Loeffler-Institute, 17493 Greifswald-Riems, Germany

**Keywords:** restricted zone, protection zone, surveillance zone, primary outbreak, secondary outbreak

## Abstract

In the event of an outbreak of African swine fever (ASF) in pig farms, the European Union (EU) legislation requires the establishment of a restricted zone, consisting of a protection zone with a radius of at least 3 km and a surveillance zone with a radius of at least 10 km around the outbreak. The main purpose of the restricted zone is to stop the spread of the disease by detecting further outbreaks. We evaluated the effectiveness and necessity of the restricted zone in the Baltic States by looking at how many secondary outbreaks were detected inside and outside the protection and surveillance zones and by what means. Secondary outbreaks are outbreaks with an epidemiological link to a primary outbreak while a primary outbreak is an outbreak that is not epidemiologically linked to any previous outbreak. From 2014 to 2023, a total of 272 outbreaks in domestic pigs were confirmed, where 263 (96.7%) were primary outbreaks and 9 (3.3%) were secondary outbreaks. Eight of the secondary outbreaks were detected by epidemiological enquiry and one by passive surveillance. Epidemiological enquiries are legally required investigations on an outbreak farm to find out when and how the virus entered the farm and to obtain information on contact farms where the ASF virus may have been spread. Of the eight secondary outbreaks detected by epidemiological investigations, six were within the protection zone, one was within the surveillance zone and one outside the restricted zone. Epidemiological investigations were therefore the most effective means of detecting secondary outbreaks, whether inside or outside the restricted zones, while active surveillance was not effective. Active surveillance are legally prescribed activities carried out by the competent authorities in the restricted zones. Furthermore, as ASF is no longer a rare and exotic disease in the EU, it could be listed as a “Category B” disease, which in turn would allow for more flexibility and “tailor-made” control measures, e.g., regarding the size of the restricted zone.

## 1. Introduction

Under the European Union (EU) legislation, African swine fever (ASF) is classified as a “Category A” disease. By definition, diseases within this category do not normally occur in the EU and immediate eradication measures must be taken as soon as they are detected [1,2]. One of these immediate measures is the establishment of a restriction zone around the outbreak farm, a proven effective animal disease control tool that has been used in the EU since the early 1980s, particularly for highly contagious diseases—such as classical swine fever (CSF) or foot and mouth disease (FMD)—that can spread rapidly from farm to farm and cause numerous secondary outbreaks [1,3].

In the case of an outbreak of ASF in domestic pigs, EU legislation requires, in addition to the stamping out of all pigs on the affected farm, the establishment of a restriction zone consisting of a protection zone with a radius of at least 3 km and a surveillance zone with a radius of at least 10 km around the outbreak [4]. The same size of zones is also required for FMD, CSF, Rinderpest, Sheep and Goat Pox, Peste des Petits Ruminants, Newcastle Disease and Highly Pathogenic Avian Influenza [4]. The purposes of the restricted zone is to prevent the spread of the disease beyond the boundaries of the restricted zone, to detect further secondary outbreaks in the vicinity of the affected farm, and to eradicate the disease within the restricted zone. In addition, the restricted zone is intended to reassure trading partners that business can safely continue as usual outside the restricted zone.

The most common measures within the restricted zone are a ban on the movement of susceptible animals, their products and other material within, from or into the zone, and active surveillance to identify other potentially infected farms.

Active surveillance measures carried out by the competent authorities include both clinical and laboratory tests, combined with censuses and biosecurity checks. If the epidemiological situation so requires, additional measures may be taken, such as the establishment of larger restricted zones or the application of much more stringent measures, e.g., preventive culling or slaughter on neighbouring farms. The competent authority may not lift the measures in the zones until a minimum period (30 days in the case of ASF) has elapsed from the date of completion of preliminary cleaning and disinfection and, where appropriate, insect and rodent control, with favourable results on the infected holding [4].

The extent of epidemics in domestic animal populations depends largely on whether or not there is farm-to-farm transmission of the pathogen. Secondary outbreaks, resulting from the transmission of the pathogen from the primary outbreak where the introduction of the pathogen first occurred, are mainly responsible for long-lasting epidemics. For example, during the major FMD epidemic in the UK in 2001, over 2000 outbreaks were recorded on farms, most of which were due to local spread following the initial introduction of FMD virus into an area [5]. The epidemic lasted 11 months (from February 2001 to January 2002) and over 6 million cattle and sheep were killed in an ultimately successful attempt to contain the disease [6]. Based on data on FMD outbreaks in Europe from 1965 to 1982, it has been estimated that a primary outbreak can lead to 20 to 40 secondary outbreaks, but in rare cases, a derailment with several thousand outbreaks can occur, as in Austria in 1973 or in the UK in 2001 [7].

Many secondary outbreaks have also been reported in CSF epidemics. In Germany, for example, 420 outbreaks of CSF were reported between 1993 and 1998, of which 93 (22%) were classified as primary outbreaks and 327 (78%) as secondary outbreaks [8]. Almost a quarter of the secondary outbreaks (24%) were in close proximity (<500 m) to the primary outbreaks.

Therefore, for a highly contagious disease such as CSF or FMD, where many secondary outbreaks may occur in the immediate vicinity of the primary outbreak farm, it is obvious and reasonable to establish restricted zones. However, experimental evidence and recent field studies have shown that ASF is a less contagious disease and that virus transmission between animals and farms is more of a delayed process [9,10,11,12,13,14,15,16,17]. As a result, initial mortality on farms remains low, although case fatality is high; over 90% of infected animals die, but usually only a few animals are initially infected [9,10,13,16,18,19]. Because of the slow transmission and low initial mortality, ASF is more akin to a category B disease [1].

If the transmission of ASF were as rapid as that of CSF or FMD or any other highly contagious disease, it would be expected to spread much more rapidly to neighbouring farms. In this context, it is appropriate to reconsider the current ASF control and eradication measures in domestic pig farms, which follow the logic of highly contagious diseases, in particular with regard to the need for large restricted zones (protection and surveillance zones).

In the case of ASF in domestic pigs, the scenarios described above become even more complicated if ASF also occurs in wild boar. Under these circumstances, it is clear that the implementation of the prescribed measures in the farms located in the protection and surveillance zones will not lead to the eradication of the disease in wild boar. Depending on the epidemiological situation, the lifting of these zones may take much longer than the minimum period required, i.e., several months, and in extreme cases, even years, which may have a significant negative impact on the pig industry.

The aim of our work was to re-evaluate the ASF outbreaks that occurred in the Baltic States (Estonia, Latvia, Lithuania) between 2014 and 2023, in particular to see how many secondary outbreaks within the protection and surveillance zones were detected by the legally required active surveillance measures. In other words, we wanted to critically examine to what extent the protection and surveillance zones have helped to detect further outbreaks in domestic pigs in the vicinity of an outbreak for which the zones were established, and to discuss whether some of the EU measures to control ASF could be modified without losing their effectiveness.

## 2. Methods

When an outbreak of ASF is confirmed, an epidemiological investigation is immediately carried out by the competent authority to identify other potentially infected holdings. Surveillance is then carried out in the restricted zone to ensure that there has been no undetected transmission of infection. The epidemiological reports of all ASF outbreaks notified in the Baltic States between 2014 and 2023 were reviewed and evaluated. These reports were prepared by the competent veterinary authority after each outbreak and contain, among other information, the results of the epidemiological enquiry and the results of the legally required active surveillance in the restricted zones. In order to gain a better understanding of the effectiveness of the surveillance measures implemented, we analysed the outbreak both spatially and over time (temporal perspective).

For the analysis, the outbreaks were classified as follows:

Primary outbreak: An outbreak that is not epidemiologically linked to any previous outbreak.

Secondary outbreak: An outbreak with an epidemiological link to a primary outbreak. A secondary outbreak can be detected either by epidemiological enquiry or by passive or active surveillance.

Index outbreak: A primary outbreak occurring during the period when no ASF protection or surveillance zone is present in any region of the country. To determine whether an outbreak farm is a new index outbreak or a new primary outbreak, we checked whether any restriction zones were present in any part of the country at the time the outbreak was confirmed. If no restriction zones were in place, it was considered to be an index outbreak (Figure 1).

The following definitions were used:

Restricted zone: Protection and surveillance zones established following the confirmation of an outbreak of ASF in a pig farm (4).

Epidemiological link: A proven link to an infected farm, either through infected animals or potentially contaminated equipment, people, vehicles, etc.

Epidemiological enquiry: An epidemiological investigation on the outbreak farm aimed at (i) determining the probable origin of the ASF virus and the means of its spread, e.g., if there is an epidemiological link with a previous outbreak; (ii) estimating the probable duration of the presence of the ASF virus; (iii) obtaining information on contact farms where the ASF virus may have been spread by animals, persons, products, vehicles, any material or other means.

Active surveillance: Legally prescribed surveillance activities carried out on domestic pig farms by the competent authorities in the protection and surveillance zones. This involves, among other things, clinical examination and collection of samples for laboratory testing.

Passive surveillance: Continuous monitoring of animals for clinical signs of ASF by farmers and farm veterinarians and reporting of suspected cases of ASF to the veterinary authorities.

Surveillance period: A period of 30 days following the primary cleansing and disinfection of an affected farm was used as the duration of the implementation of surveillance measures in the restricted zone, which is the minimum duration of restrictions in a surveillance zone according to the EU legislation (4).

For every outbreak farm, the date of confirmation and the distance from the primary outbreak were identified. In case of outbreaks following the index outbreak but were also new primary outbreaks, the distance from the nearest previous outbreak in time was identified.

Summary statistics were performed on the collected data using XLSTAT 2022.2.1. MS Excel was used to generate the graphs.

## 3. Results

From 2014 to 2023, a total of 272 outbreaks in domestic pigs were confirmed in the Baltic States: 29 in Estonia, 83 in Latvia and 160 in Lithuania (Table 1). Estonia reported 8 outbreaks on backyard farms, Latvia 65 and Lithuania 141; the remaining outbreaks occurred on commercial farms. Of the 272 outbreaks, 263 (96.7%) were primary outbreaks and nine (3.3%) were secondary outbreaks. None of the primary outbreaks had more than one secondary outbreak and none of the secondary outbreaks became the source of a tertiary outbreak. None of the secondary outbreaks were detected by active surveillance. Eight of the secondary outbreaks were detected by epidemiological enquiry and one by passive surveillance. Of the eight outbreaks detected by epidemiological enquiry, six were within the protection zone, one was within the surveillance zone and one outside the restricted zone, 33 km away from the primary outbreak (Table 1 and Figure 2).

As far as the primary outbreaks are concerned, all of them were detected by passive surveillance, except for one outbreak in Latvia, which was detected by active surveillance.

The mean distance of secondary outbreaks from primary outbreaks was 5.3 km, with a median of 0.8 km (minimum 0.02 km and maximum 33 km) (Figure 3).

The secondary outbreaks were detected within one to 24 days after the respective primary outbreak (mean 8.9 days, SD 6.7) (Table 2).

Data on the distribution of outbreaks between zones detected after the index case by country in different years are presented in Table 2. In total, 14.9% (*n* = 37) of all outbreaks following the index outbreak in all years and countries were detected within one of the established protection zone, 11.7% (*n* = 29) in one of the established surveillance zone and 73.4% (*n* = 246) outside the borders of any existing surveillance zones. The proportions observed varied between years and countries (Figure 4 and Table 2). The detection rate of outbreaks within existing restricted zones was the highest in 2014, when 72.2% of all outbreaks were detected in protection (52.8%) or surveillance zones (19.4%). However, this was strongly influenced by the situation in Latvia in 2014, where several outbreaks were detected in closely located smallholder pig farms. Furthermore, the protection and surveillance zones established in 2014 in Latvia exceeded the minimum radii of 3 km and 10 km, respectively, and were set along the boundaries of administrative units of the affected area. Since 2015, the detection rate of outbreaks within existing restriction zones has been below 50%.

## 4. Discussion

Ideally, from a disease management perspective, with effective disease control and good biosecurity on farms, there should be no secondary outbreaks. However, many secondary outbreaks do occur, especially in epidemics of highly contagious diseases. Experience with FMD and CSF shows this [5,8], but in the case of ASF, which is a low-contagious disease [9,13,16], secondary outbreaks are expected to be few if early detection is in place. Our results support this, as the vast majority (96.7%) of outbreaks in the Baltic States were primary outbreaks. The potential epidemiological links between outbreaks were examined during the epidemiological investigations by careful examination of all incoming movements to the farm (animals, vehicles, goods and people) during the period of the most probable time of virus introduction and could not be established on the farms classified as primary outbreaks, but could be established for the secondary outbreaks. Due to the complex structure of the ASF virus, molecular tracing of virus strains is not yet feasible in epidemiological investigations. Furthermore, the methods currently available do not allow for the differentiation of farm-specific virus strains. In situations where the virus is also circulating in the wild boar population, the detection of the same virus strain in two pig herds does not immediately imply that there is an epidemiological link between the herds.

The relatively low number of secondary outbreaks also shows that ASF did not spread rapidly from one domestic pig farm to another but was introduced from sources not linked to other outbreaks. One could speculate that the ASF virus entered the majority of farms from a contaminated environment via contaminated fomites, feed, vehicles, etc. However, there is no clear evidence to support this assumption. Nevertheless, it is recommended that biosecurity measures should be improved on all pig farms to reduce the risk of introduction. Although the primary purpose of restricted zones is to prevent the spread of disease due to the standstill measures, they should also facilitate the detection of further outbreaks within the zone through the legally foreseen active surveillance. However, the secondary outbreaks were mainly detected by epidemiological enquiry and the surveillance measures in the zones did not play a major role in their detection, emphasising that effective epidemiological enquiry is crucial for the detection of secondary outbreaks. Only one outbreak was detected by active surveillance, which turned out to be a primary outbreak not linked to other outbreaks, although the outbreak was located within a restricted zone.

Eight out of nine secondary outbreaks were detected in protection or surveillance zones and the majority of these were within one kilometre of the primary outbreak. The secondary outbreaks were detected on average one week after the primary outbreak was confirmed. It can be assumed that the movement restrictions during this period may have prevented further spread of the disease from these farms.

In summary, the role of active surveillance, which is legally required in restriction zones, has not played a major role in the detection of secondary outbreaks. Nevertheless, the compulsory farm visits by the veterinary authorities may be important in raising awareness and motivating farmers to improve biosecurity by reminding them of the requirements and restrictions in the restricted zones. This may have an indirect effect on reducing further outbreaks. Effective biosecurity measures are necessary to prevent the introduction of the ASF virus, e.g., from the wild boar habitat or from other sources, in particular those related to human activities. Furthermore, account must be taken of the fact that the disease is not always recognized promptly, and it may take some time before epidemiological studies can reconstruct the dynamics of the infection and the farm-to-farm transmissions can occur before the primary outbreak was detected and restrictions implied as it has been observed during the FMD or CSF epidemics. Our findings, however, demonstrate that in the case of ASF, this does not occur frequently. Nevertheless, the establishment of restriction zones is the only immediate tool available to block transmission, especially in the context of densely populated areas or large smallholder sectors.

A fundamental requirement of disease management for any transmissible animal disease is that prevention and control measures should be “tailor-made” and disease-specific to address its unique epidemiological profile, consequences and distribution within regions or countries [16]. In the case of ASF, this would mean that the low level of contagiousness should be taken into account in the design of control measures, in particular in the establishment of restricted zones. However, the sizes of the restricted zones for ASF have been taken over from the previous CSF legislation [20,21]. As a result, the control of ASF largely follows the measures taken to control CSF, i.e., a less contagious disease is controlled along the lines of a highly contagious disease. As a result, areas with a radius of at least 10 km may be subject to restrictions for several months, even though the likelihood of secondary or new primary outbreaks is low. This is not only an enormous burden for all the pig farms affected, but also for the veterinary authorities, who have to carry out laborious and costly control and surveillance measures, the effectiveness of which appears to be low in relation to the effort and resources expended.

A recent EFSA scientific opinion [22] concluded that the sizes of the protection and surveillance zones for ASF were highly effective, but that hypothetical protection and surveillance zones of 2 and 4 km radii were also found to be effective. An example of a more flexible and smaller restriction zone can also be found in the United States Department of Agriculture (USDA) ASF Preparedness and Response Plan [23], which recommends an “infection zone” of at least 3 km, plus a “buffer zone” of at least 2 km beyond the infection zone. Based on our experiences, we do support smaller restricted zones as suggested in the EFSA opinion.

The reduction of the radius could be decided by the national veterinary authorities on the basis of the location of the outbreak in areas where restrictions are already in place due to the occurrence of ASF in wild boar, the local farming systems and the density of pigs and farms, especially backyard farms.

In the EU, outbreaks of ASF in domestic pigs often occur in a zone that is already restricted due to the ongoing epidemic in the wild boar population (the so-called “Restricted Zone II”) [24]. In such a case, the need for the establishment of the additional restricted zone (protection and surveillance zones) following ASF in domestic pigs is questionable, as the ASF outbreak has occurred in an area where restrictive movement and trade measures for domestic pigs are already in place. In these circumstances, there is therefore a further reason to reduce or adapt the size of the protection and surveillance zones for domestic pigs. The overlapping of too many measures does not help the authorities to prioritise and use resources effectively.

ASF is currently classified in the EU as a “Category A” disease, which requires immediate eradication measures to be taken as soon as it is detected. Category A diseases are those that are considered to be highly transmissible and do not normally occur in the EU [1,2]. However, neither of these two basic criteria apply to ASF. The disease is not highly transmissible in either domestic pigs or wild boar [9,13,16]. Furthermore, it can no longer be considered exotic to the EU, as 13 EU countries are currently affected by ASF (mainly in wild boar), and most of them have been for many years [17,25,26,27]. For example, ASF has been regularly detected in the Baltic States since 2014 [25,27].

Finally, it can be concluded that the possibility to reduce, or, respectively, to adapt the size of the protection and surveillance zones according to the different epidemiological circumstances should be foreseen in the EU legislation in order to allow for more “tailor-made” control measures, especially in case of outbreaks in domestic pigs in areas where restriction zones are already in place due to ASF in wild boar. More generally, a discussion should be launched on the possibility of revising the categorisation of ASF, as more and more evidence proving that ASF fulfils the criteria of a category B disease rather than a category A disease is emerging. Nevertheless, even as a category B disease, ASF must be controlled in all Member States with the aim of eradicating it throughout the Union [1].

## Figures and Tables

**Figure 1 viruses-16-00334-f001:**
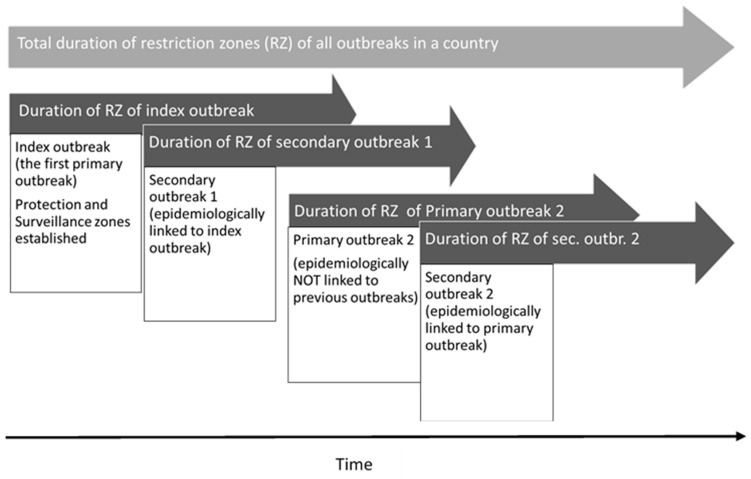
Duration of restriction zones in relation to the different types of outbreaks (index, primary and secondary).

**Figure 2 viruses-16-00334-f002:**
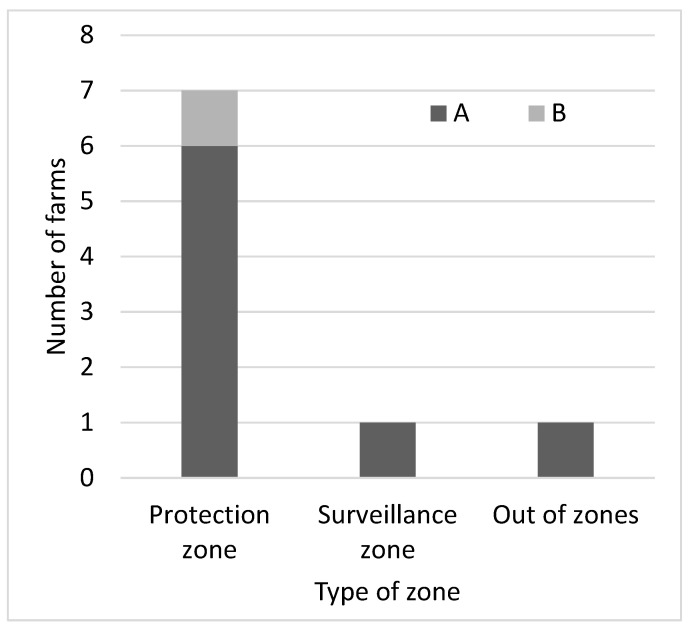
Mode of detection of secondary outbreaks of ASF in the Baltic States from 2014 to 2023 (A—by epidemiological enquiry; B—by passive surveillance).

**Figure 3 viruses-16-00334-f003:**
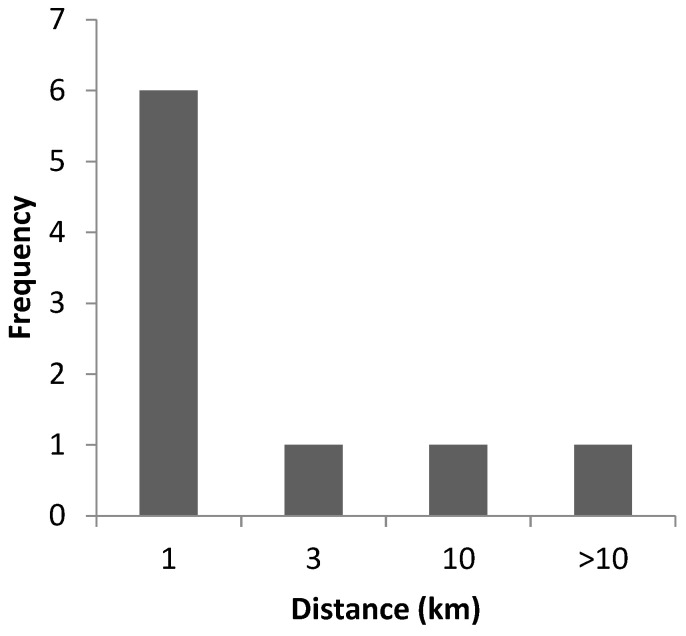
Distances of secondary ASF outbreaks from primary outbreaks in the Baltic States from 2014 to 2023.

**Figure 4 viruses-16-00334-f004:**
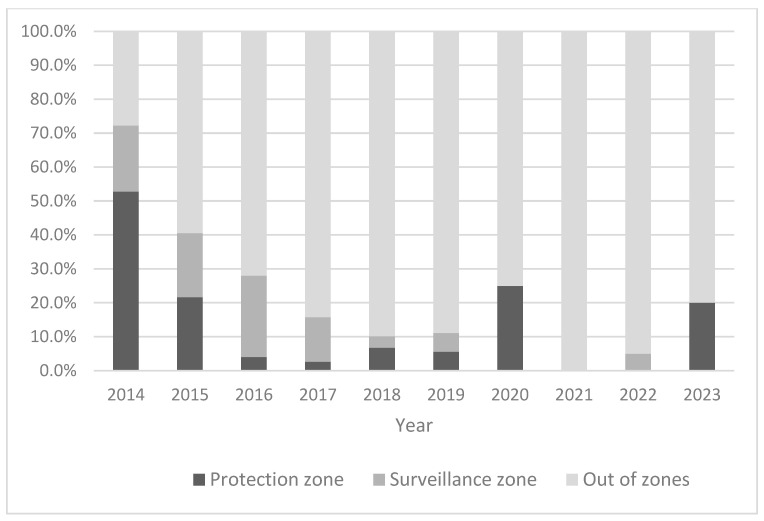
Detection rate of outbreaks following the index outbreak in different restriction zones in the Baltic countries from 2014 to 2023.

**Table 1 viruses-16-00334-t001:** Overview of the ASF outbreaks in the Baltic States from 2014 to 2023.

Year	Estonia		Latvia		Lithuania		Total
	Prim	Sec	Prim	Sec	Prim	Sec	
2014	-	-	28	4	6	-	38
2015	15	2	10	-	13	-	40
2016	6	-	3	-	19	-	28
2017	3	-	7	1	30	-	41
2018	-	-	10	-	49	2	61
2019	-	-	1	-	19	-	20
2020	-	-	3	-	3	-	6
2021	1	-	2	-	0	-	3
2022	-	-	6	-	16	-	22
10/2023	2	-	8	-	3	-	13
Total	27	2	78	5	158	2	272
	29		83		160		

**Table 2 viruses-16-00334-t002:** Type of secondary outbreaks, time period and distance from primary outbreaks.

Country	Days between Detection of Primary and Secondary Outbreaks	Distance (km)	Type
Estonia	1	9.91	A
Estonia	11	0.02	B
Latvia	7	0.78	A
Latvia	11	1.05	A
Latvia	11	0.97	A
Latvia	9	0.51	A
Latvia	24	33	A
Lithuania	1	0.79	A
Lithuania	7	0.71	A

A: Outbreak detected by epidemiological enquiry. B: Outbreak detected by passive surveillance.

## Data Availability

No new data were created.

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
