# Peer review of "The Effectiveness of Protection and Surveillance Zones in Detecting Further African Swine Fever Outbreaks in Domestic Pigs—Experience of the Baltic States"

_viruses, 2024, doi:10.3390/v16030334_

Round 1
Reviewer 1 Report (Previous Reviewer 2)
Comments and Suggestions for Authors
Thank you for your work and its contribution to managing highly-contagious disease. Zone size has an arbitrariness that comes from committee aversion to risk.
Author Response
Many thanks for the very positive and constructive feedback. As there were no further comments, we have left the text as it is.
Reviewer 2 Report (Previous Reviewer 1)
Comments and Suggestions for Authors
Compared to the previous review, the methodological part of the study and the critical evaluation of the results have not improved. The data does not support authors' considerations.
The authors, also as reported in the second response, base the outcome of their study on the basis of 9 secondary outbreaks, an irrelevant number (3.3%) compared to the total number of outbreaks.
The data they present can also lead to completely different interpretations from those proposed by the authors. For example, have the authors critically evaluated the fact that 50% of primary outbreaks still fall within the protection and surveillance area? Therefore, this could imply that the application of restrictive measures have played a role in the further spread of the infection from those areas? which is the meaning of the application of measures in these areas. The number (50%) is not irrelevant and justify the application of such measures. Especially if the origin of the infection in the farms has not been identified.
It is important to point out that in the previous review there was no criticism of the activities of the veterinary services, the criticism is on the way the data deriving from their activities are used in this study.
Author Response
We are of exactly the opposite opinion regarding the small number of secondary outbreaks (3.3%). We see the most relevant aspect of our study in the fact that there are so few secondary outbreaks. This finding makes a very significant statement about the nature and epidemiology of ASF compared to the other Category A diseases. And that is exactly what this study is about.
As mentioned in the previous responses to the reviewer, this paper is NOT about primary outbreaks, it is not about analysing primary outbreaks, which is clearly stated in the introduction. The study was only concerned with how many secondary outbreaks were identified by the authorities after the implementation of the legally required measures.
As for the reviewer's comment that his remarks should not be understood as criticism of the veterinary services, we just quote his comment from his first review, where he questions the competence of the veterinary authorities: "Were the investigations conducted by experienced veterinarians?" Such comments are neither constructive nor collegial.
If the outbreak investigations had not been carried out by competent and experienced colleagues, we would not have had to conduct the study at all.
Reviewer 3 Report (New Reviewer)
Comments and Suggestions for Authors
Because the manuscript was already reviewed, I do not have any significant questions to this paper. Also we need to accept that the paper written by real world top field expert in ASF control and epidemiology.
This paper is really important to stimulate the discussions on legislation adaptation to the current epidemiological situation with ASF in Europe.
The limitation of this paper - it is not about ASF in general, it is about the disease caused by Georgia-2007 ASFV strain. May be we need to follow the Avian Influenza experience and divide control measures for HP and LP strains of ASF. I do not know will we have the same scenario with NH/P68 ASFV strain if they will be introduced in EU, as it already happened in China.
In conclusion in any cases the paper can be accepted for publication and it is important to publish it as soon as possible.
Author Response
Many thanks for the very positive and constructive feedback. We also hope that our manuscript will serve as a stimulus for further discussion, and that new proposals for better solutions will follow.
Reviewer 4 Report (New Reviewer)
Comments and Suggestions for Authors
I appreciate the great manuscript crafted by Lamberga et al. and am honored to review it. The reported work is definitely appealing and valuable to better understand the transmission and control of ASF. However, I found critical information lacking and issues in readability. Please find my comments in the PDF. Thank you.

Author Response
All comments from the reviewer are very welcome and will definitely help to improve the manuscript. We found the comments to be very constructive and helpful. We were able to take all points into account and make appropriate improvements.
Abstract: The definitions of primary outbreaks, secondary outbreaks and epidemiological enquiry were included. We have amended the text as suggested by the reviewer by including the results of how many secondary outbreaks have been detected in the restricted zone and by what means. The detection of secondary outbreaks of ASF is mainly due to epidemiological investigations rather than active surveillance. Therefore, the size of the restricted zone does not play a major role in the detection of secondary outbreaks.
Introduction: Clear definitions of primary and secondary outbreaks are under Materials and Methods. However, we have amended the text to make it clearer.
Methods: The suggestions concerning the secondary outbreaks were followed. For more clarity we introduced a new Figure which illustrates the types of the outbreaks (primary, index and secondary).
Results: The critical findings were included in the abstract. To avoid confusions and misunderstandings we have deleted the text highlighted by the Reviewer concerning the data on distribution of outbreaks between the zones as well as Figure 4. Table 2 has been amended accordingly. The sections of text that appeared unclear have been improved.
Discussion: In the first comment of discussions, the reviewer is arguing: “The few secondary cases for ASF can be caused by undetected/undetermined links between primary and secondary outbreaks sites.” There is a contradiction in this argument. When the connections remain undetected/undetermined, it cannot be a secondary outbreak. By definition, secondary outbreaks are only those that can be traced back to a primary outbreak. If this is not the case, the outbreak is considered primary. However, the point made by the reviewer that ASFV can remain infectious in the environment, e.g. in fomites, and spread between personnel/materials off-site is valid and we have included it in the text.
The reviewer's point about the wild boar habitat and farm biosecurity measures never being an objective is valid. To avoid confusion, we have removed these sections.
The point maid by the reviewer that restricted zones aim to prevent the spread of disease, not for case detection, is valid. We have changed the text accordingly.
The reviewer did not agree with our conclusion which states: “Our data suggest that in the case of ASF, the size of the protection and surveillance zones could be reduced, e.g. to 3 km.” We deleted the sentence.
The last point made by the reviewer that no risk assessment has been made is correct, we have deleted that sentence.

Round 2
Reviewer 4 Report (New Reviewer)
Comments and Suggestions for Authors
I appreciate the authors for taking my comments into account and improving the manuscript in many aspects.
This manuscript is a resubmission of an earlier submission. The following is a list of the peer review reports and author responses from that submission.
Round 1
Reviewer 1 Report
Comments and Suggestions for Authors
see attachment

Author Response
All comments from the reviewer are very welcome and will definitely help to improve the manuscript. We found the comments to be very constructive and helpful. We were able to take all points into account and make appropriate improvements.
Methodology When carrying out the investigation, did you try to evaluate how long the disease had been present on the farm? This could be useful in establishing the chain of transmission, especially when almost all outbreaks were primary, as reported. In primary cases, was it possible to at least formulate a possible route of introduction of the infection? Even if not certain. It would be useful to know how many of the primary outbreaks were on commercial farms and how many in backyards.
The outbreak investigations in the three Baltic States were carried out by the competent authorities, mostly the local veterinary authorities. We have evaluated these reports. Precise information on the period of high risk was not always available. This would have been very helpful information, but unfortunately it was not always available. As for the information on outbreaks in commercial farms versus backyard farms, we can provide this information and have included it in the text. There were 8 outbreaks in backyard farms in Estonia, 65 in Latvia and 141 in Lithuania.
Results/Considerations: I would suggest you also make some considerations on the fact that the disease is not always recognized promptly and that it may be necessary to have some time available before being able to reconstruct the dynamics of transmission of the infection by carrying out epidemiological investigations. Establishing restriction zones is often the only immediately available tool to block transmission, especially in certain livestock contexts, densely populated areas (where there are also many exchanges) or backyard setings (biosecurity, under reporting ..)
The study is largely built on the results of the epidemiological investigation, given the number of primary outbreaks some considerations on the quality of the investigations and/or tracing system is suggested: detected deficiencies, lack of transparency on the part of the operators .. Were the investigations conducted by experienced veterinarians? Some considerations on biosecurity applied in outbreaks would also be relevant, given the possible role of wild boars. Furthermore, it is worth considering that in certain livestock contexts (dense areas) the establishment of protection and surveillance zones for ASF is more meaningful than it is for diseases such as highly pathogenic avian influenza or FMD, whose transmission is also supported by population density, sometimes alone. I am reporting these considerations to suggest better evaluating the results of the study using a broader look. Indeed, the legislation should adapt to different livestock contexts which are not always comparable to the reality of the Baltic countries.
The comment that the disease is not always recognized promptly is very true, and it may take some time before epidemiological studies can reconstruct the dynamics of infection transmission. We have included this comment in the discussion, as well as the comment on the establishment of restriction zones in the context of densely populated areas. The following text has been added: “Furthermore, account must be taken of the fact that the disease is not always recognized promptly, and it may take some time before epidemiological studies can reconstruct the dynamics of the infection. The establishment of restriction zones is the only immediate tool available to block transmission, especially in the context of densely populated areas. The epidemiological situation in the Baltic States is not necessarily comparable with other infected areas in Europe, e.g. with a much higher livestock density.”
On the question of whether the investigations were carried out by experienced veterinarians, the only thing that can really be said is that all the veterinarians involved have gained a lot of experience over the years. The knowledge about ASF in 2014 was certainly not comparable to that in 2023. This applies not only to the veterinarians who carried out the investigations in the field but also to the authors of this paper. Therefore, a certain subjectivity cannot be ruled out.
Line 232-234: this sentence should be supported by the outcomes of the investigations, in “results”. Given the objective of the study, the results are greatly influenced by the high number of primary outbreaks, I believe that the analysis should also focus on this. Indeed, the difficulty of tracing could justify the presence of large restriction areas which does not necessarily mean that they must be the protection and surveillance zone, they could be zone III.
In fact, the high number of primary outbreaks influences the interpretation of the results. This high number of primary outbreaks was neither expected nor anticipated before the start of the study. We will focus on this finding in a follow-up study, particularly when the biosecurity factor on pig farms is discussed.

Reviewer 2 Report
Comments and Suggestions for Authors
Very interesting and well-written paper. Excellent contribution to ASFv response planning. One suggestion: the primary focus of the paper, "to see how many secondary outbreaks within the protection and surveillance zones were detected." This makes Table 2, with distance, focal data. The suggestion to reduce the size of zones is done without statistical analysis. Why? Were 8 observations too few? What analysis was done to suggest “e.g. to 3 km” (line 262-263)? A fuller discussion of 'zone size', utilizing reference #22, would support the primary purpose of this work.
Author Response
The reviewer's comment is very welcome and will certainly help to improve the manuscript.
Very interesting and well-written paper. Excellent contribution to ASFv response planning. One suggestion: the primary focus of the paper, "to see how many secondary outbreaks within the protection and surveillance zones were detected." This makes Table 2, with distance, focal data. The suggestion to reduce the size of zones is done without statistical analysis. Why? Were 8 observations too few? What analysis was done to suggest “e.g. to 3 km” (line 262-263)? A fuller discussion of 'zone size', utilizing reference #22, would support the primary purpose of this work
We have taken up the reviewer's comment and add to the discussion a short text with a clearer reference to the EFSA study (Reference #22).
Round 2
Reviewer 1 Report
Comments and Suggestions for Authors
See attachment

Author Response
Line 52-60: The main aim of the restriction zone (PZ. SZ) is to contain/limit the spread of the disease. This is the reason why movements are limited. In relation to protection and surveillance zone, see REGULATION (EU) 2016/429:
The text has been improved according to the reviewer's suggestion: “ The purposes of the restricted zone is to prevent the spread of the disease beyond the boundaries of the restricted zone, to detect further secondary outbreaks in the vicinity of the affected farm, and to eradicate the disease completely within the restricted zone so that farming activities and trade can resume as soon as possible.”
Then, when the outbreak is confirmed, the investigation is immediately carried out to identify other potentially infected farms. Afterwards, surveillance in the protection and surveillance zones is to verify that there has not been an unidentified transmission of the infection and in fact, it is preliminary to the release of the measures. In this light should be revised the sentence reported in paragraph 109-113 and in the discussion as well. Indeed, the epidemiological investogation is main tool to trace the infection.
The text was supplemented accordingly as suggested by the reviewer: “When an outbreak of ASF is confirmed, an epidemiological investigation is immediately carried out by the competent authority to identify other potentially infected holdings. Surveillance is then carried out in the restricted zone to ensure that there has been no undetected transmission of infection.”
In the study, it is necessary to improve the analysis on the primary outbreaks. Where were all the primary outbreaks located? Was an analysis carried out with respect to the cases in wild boars? Is it possible that no reference is made to the role of swill feeding given that it is essentially backyard setting?
This study is not about analyzing primary outbreaks, which is clearly stated in the introduction. It is also irrelevant where these primary outbreaks occurred (by the way, they were spread over the entire Baltic region). The study was only concerned with how many secondary outbreaks were identified by the authorities after the implementation of the legally prescribed measures. It is in no way our place to retrospectively scrutinise the work of the responsible authorities involved in the investigations, for example whether they carried out investigations with respect to the cases in wild boar. We were confident and fully trust that the authorities worked conscientiously and that they naturally took the wild boar cases into account. Throughout the years of the epidemic, the veterinary authorities of the Baltic states have distinguished themselves by their conscientiousness and professionalism in combating the disease. At least we are not aware of anything to the contrary.
The comment regarding swill feeding is also not relevant in this context, as our work does not examine the pathways by which the pathogen is introduced into a (primary outbreak) farm. Our work is primarily only concerned with how many secondary outbreaks have followed a primary outbreak. Swill feeding does not usually lead to secondary outbreaks.
Line 212-214: considering the purpose of the study, this sentence must be supported, the spread of 97% of outbreaks cannot be dismissed in this way, given that you are talking about a revision of control measures claiming that are not effective for ASF while, from what appears from these data, is the investigations that have failed.
The investigations did not fail, but the opposite was the case. They were able to clarify how many outbreaks were primary and how many outbreaks were secondary. Rather, the investigations have revealed that the introduction of the pathogen into a farm is not efficiently prevented, e.g. due to insufficient biosecurity measures resulting in a high percentage of primary outbreaks. To make this even clearer, we have added the following sentence. “The findings strongly suggest that the introduction of the pathogen into a farm is not effectively prevented and that this could explain the high percentage of primary outbreaks. It is a clear indication that biosecurity measures on farms need to be improved It is a clear indication that biosecurity measures on farms need to be improved, particularly in regions where wild boar are affected.”
Line 216 – 217: it is true but what is said in this paragraph does not seem in line with the results of this study. This study is undermined by the results of the investigations which in 97% of the outbreaks, apparently, produced no results.
Exactly the opposite is the case, the study has indeed produced results. The study has shown that there were only a few secondary outbreaks (3%) during the ASF epidemic in the Baltic states. And this is precisely the result of the study. The study did not investigate the question of why there were proportionally so many primary outbreaks (97%), although this question can also be answered: Lack of biosecurity.
